# An experiment on the impacts of experiential investment advice

Garret Ridinger[1], James Sundali[1]*, Federico Guerrero[1], Mauricio Solorio[2],
Mengyue Fan[1], Irem Sevindik[1], Qifan Chen[3], Diana Neely[1]

1 College of Business, University of Nevada, Reno Reno, Nevada, United States, 2 State of Nevada, Senior Economist, Carson city, Nevada, United States, 3 College of Business and Economics, Longwood University, Farmville, Virginia United States

* jsundali@unr.edu

## Abstract

An experiment is conducted in which investment advice is passed from one cohort to the next. Participants make asset allocation decisions for thirty years to a safe and risky asset and provide annual forecasts (beliefs) on the return on the risky asset. Risky asset returns are drawn from the price returns on the S&P 500 from 1921–2010; Cohorts 1/2/3 received the actual returns in the time period from 1921–1950/1951–1980/1981–2010 respectively. Results show that *negative* investment advice passed from Cohort 1 to Cohort 2 leads to: 1) significantly lower allocations to the risky asset in Cohort 2 compared to Cohort 1; and 2) a 19% difference in allocations between cohorts who received either positive advice or negative advice in Cohort 2. A second experiment examines the effect on Cohort 3 from receiving consistent and mixed advice from Cohorts 1 and 2. The results from the first experiment are replicated showing that positive (negative) advice received from prior cohorts leads to higher (lower) investment beliefs and portfolio allocations to a riskier asset. Statistical analyses are conducted to determine if the large differences in allocations are driven by changes in beliefs about future returns or changes in risk attitude. The primary takeaway is that experiential cohort advice has a significant impact on subsequent cohorts even when it has little informational value.

## Introduction

How do investors form beliefs about future asset returns and determine how much risk they are willing to accept? Imagine a young and naïve investor is presented with the annual returns on the S&P 500 from 1872–2022 [1] as shown in Fig 1. The mean of this return stream is 6% and the standard deviation is 18%. After studying this graph, will this investor assume that the returns she will experience over the next thirty to forty years will be equal to 6% or will she expect a severe market crash like the one from 1930 to 1933? Will this investor be willing to accept the ups and downs

**Data availability statement:** Data available at Science Data Bank: https://www.scidb.cn/en/anonymous/SXZJN052.

**Funding:** The author(s) received no specific funding for this work.

**Competing interests:** NO authors have competing interests.

in market returns and her portfolio balance or would she be more comfortable in a stable but lower return asset? Will this investor be able to answer these questions before she begins her investment journey, or will she only be able to answer these questions with experience?

Numerous studies have found wide variation in stock market participation [2–4], portfolio allocations [5,6], and beliefs about asset returns [7,8]. While numerous factors have been shown to influence this heterogeneity, it remains an open question as to where these variations in investor behavior come from. Research by Brau et. al. [9] suggests that financial literacy among college students is higher when it comes from experiential learning activities compared to family and background influences or from formal learning activities. The focus of the research presented here concerns the impact of experiential peer advice, often referred to as "generational advice" in the experimental literature [10,11]]. We will use the term "generational advice" to be consistent with prior research and to make it clear that advice is coming from a prior experienced cohort. We are not suggesting this equates to actual advice passed between family generations (i.e., parent to child). Prior field studies [12] and lab studies [13] have shown significant peer effects on financial decision making and suggest the impacts could be due to social learning, social utility, or conceptual learning. Our study evaluates whether the framing of advice, either positive or negative advice, passed from one experienced peer cohort to the next will impact investment behavior. The analogy we use to describe our experiments is whether the beliefs and actions of one generation can be influenced by a prior generation.

Experienced investors may pass on their own market experiences, risk preferences, or beliefs about investing that can influence future investment behavior of subsequent investors The academic financial literature generally suggests there are two channels through which prior advice may impact investment decisions: 1) either by impacting beliefs about the future returns on risky assets (beliefs); or 2) by impacting an individual's tolerance for risk (risk aversion) [14,15].

In this paper we experimentally explore how the advice of one cohort can affect the investment allocations and beliefs of the next cohort. We present the results from two experiments, both with the following basic design. Participants are given money to allocate between a safe and risky asset for thirty periods. Participants make allocation decisions and provide information regarding their investment beliefs and risk attitudes. Investment beliefs are measured as the participant's forecast of the return on the risky asset in the next period. The participant's risk attitude regarding the risky asset is measured from survey questions answered by the participant once every ten periods. The experimental manipulation is the "experiential advice" information given to participants prior to making their investment decisions. In Experiment 1, a control group (Generation 1) begins by making their allocation decisions without receiving any advice. At the conclusion of the experimental task, Generation 1 is asked to provide information to pass along to the next cohort (Generation 2) that will participate in the experiment. The information is framed to Generation 2 as if they are receiving investment advice from their parents. There are two differing treatments in Generation 2, one cohort receives positive advice from Generation 1, and a second cohort

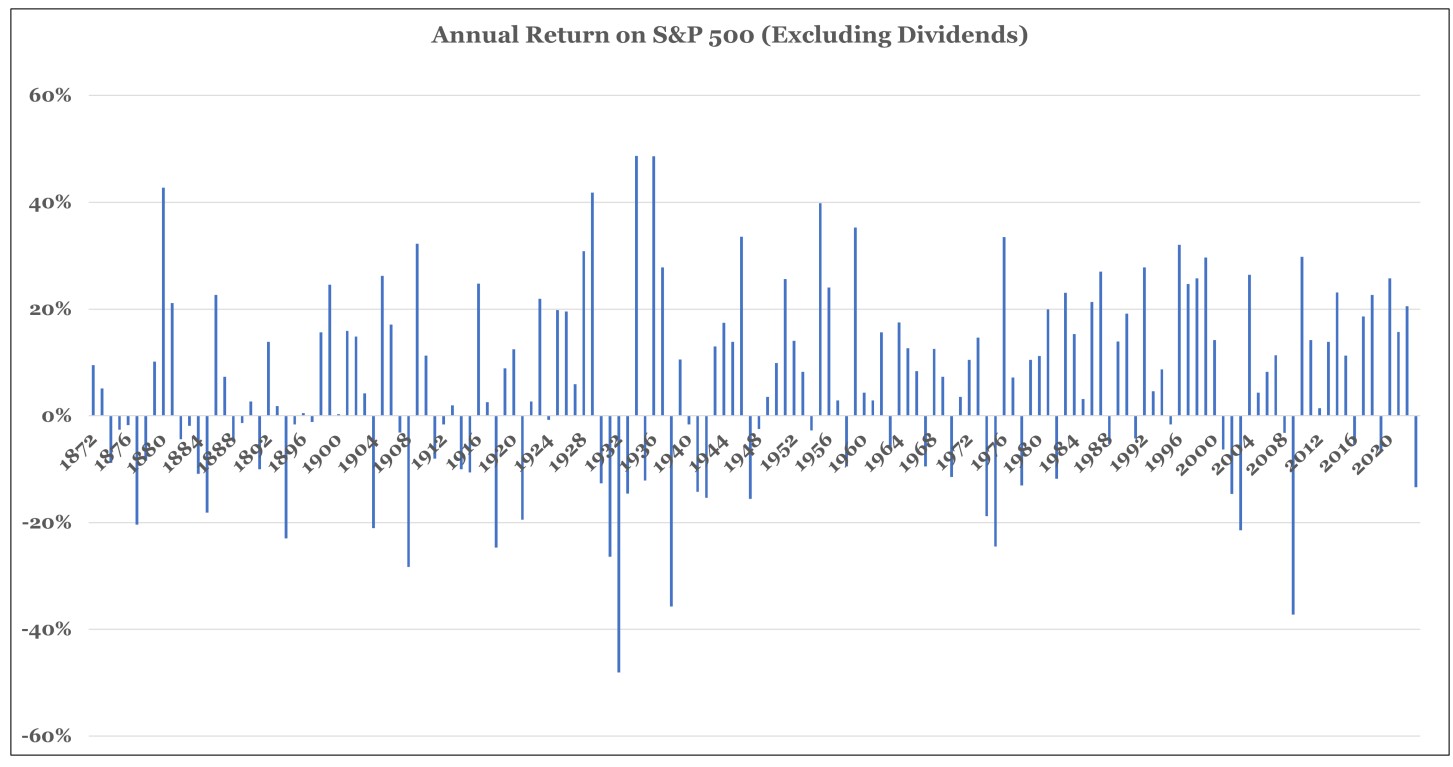

**Fig 1. Annual returns on S&P 500.**

receives negative advice. The experimental results show a quantitatively large and statistically significant difference in the allocation decisions and beliefs between the positive and negative advice cohorts of Generation 2. Participants receiving positive advice form more positive beliefs and make significantly higher investment allocations to a risky asset.

Experiment 2 is designed to replicate and extend the results of Experiment 1. The participants in Experiment 2 are dubbed Generation 3 and receive advice from both their parents (Generation 2) and grandparents (Generation 1). There are four experimental conditions in Experiment 2 depending on the mix of advice passed from Generation 1 and Generation 2. The advice passed to Generation 3 from Generations 1 and 2 is framed as Positive/Positive, Positive/Negative, Negative/Positive, and Negative/Negative, respectively. Once again, the experimental results show that participants receiving Positive/Positive advice from Generations 1 and 2 allocate more to the risky asset and have more positive investment beliefs. Our results clearly show that the positive or negative framing of intergenerational advice has a quantitively large and statistically significant impact on allocations to a risky asset. Our statistical analyses explore whether these large differences in allocations can best be tied causally to changes in risk attitudes and/or changes in investment beliefs.

The rest of the paper is organized as follows. We begin with a brief review of the relevant literature. We then present the design of an investment experiment and present our hypotheses. Following analysis of the results we present our conclusions and directions for future research.

## Literature review

### Social learning and advice literature

The social learning literature in economics and finance examines how people learn from other individuals when they are making decisions to invest [16]. This type of social learning (or peer effects) occurs when people learn by observing

behavior or information learned from others such as fellow market participants and market experts [12,17]. An investor may assume that a peer has relevant private information and chooses to mimic the behavior which can lead to informational cascades and/or herd behavior [18,19]. Social norms [20] encourage investors to follow the crowd because of desires for conformity, social sanctions, identity considerations or strategic complementarities [17,21].

A study by Schotter and Sopher [10] examines "intergenerational games, i.e., games played by a sequence of non-overlapping agents, who pass advice on how to play the game across adjacent generations of players" (pg. 123). They found advice can have a significant influence on subsequent individual behavior in strategic interactions. Specifically, they find that intergenerational advice in the trust game decreases trust in senders but increases reciprocity in responders. Research studying advice in intergenerational public good experiments has found that cooperative advice can increase cooperation by the next generation, but this effect is strongest in the beginning of the game [22]. In studying dynamic intergenerational public good games, Sherstyuk et. al. [11] find advice can help groups achieve greater dynamic efficiency but can also lead groups to converge on noncooperative outcomes. In an experiment on social learning Celen et al. [23] found a stronger effect of advice from prior cohorts on individual behavior compared to observing the prior cohort's behavior.

A large amount of literature has focused on studying the influence of advice that financial advisors give to their clients. Foerster et al., [24] show that even after controlling for risk tolerance, time horizon, and expertise, a financial advisor's own portfolio is strongly predictive of the investments chosen for the client. High levels of trust in an advisor is key in explaining whether individuals engage with expert advice. Gurun et. al., [25] shows that broken trust due to fraud exposures can spread through social networks leading to withdrawal of funds with advisors and increase of deposits in safer assets. In contrast, Pogrebna [26] finds no evidence that individuals making high stakes monetary decisions in a television show are influenced by advice received by the audience.

A study closely related to ours is Alevy and Price [27], who examine the role of intergenerational advice on experimental asset markets. Again, "intergenerational advice" is a proxy term for advice passed from one experienced cohort the next. In the experiment, subjects participated in multiple periods of trading assets that pay a random dividend with common knowledge of the dividend distribution. In the design, future generations participated in the same underlying trading environment, but some individuals received advice from a prior participant. The results of the experiment showed that advice served as a substitute for experience leading to prices shifting towards the fundamental value. Similar to Alevy and Price [27], we allowed subjects to submit unrestricted advice to the next generation. However, in our design, we varied whether the next generation received positive or negative advice from the prior cohorts. In an asset market experiment, the other participants' decisions can influence the prices in the experiment. As a result, advice may help individuals better coordinate on the reaching the fundamental value. This makes it difficult to tease out whether advice helps individuals better understand the market dynamics or if it helps them reduce strategic uncertainty. In our setting, individual portfolio choices do not impact others. As result, our study directly examines whether intergenerational advice influences non-strategic investment decision making.

## Experiment 1: Task, research design, participants, hypotheses

### Methods

**Participants.** 117 participants took part in Experiment 1. Participants were recruited from announcements in classes and a university wide subject pool and signed up using the SONA participant pool software. Subject recruitment for all of the experiments reported in this research began on April 14, 2022 and ended on November 7, 2022. This experiment, and all subsequent experiments reported in this paper, began with participants reading and signing an informed consent form that was approved by the University of Nevada, Reno Social Behavioral Institutional Review Board (Protocol # 1128451−1) (see Appendix I in S1 File for consent form) (IRB approval was granted on 10/06/2017 and reaffirmed on 10/26/2021). The total number of participants in Experiment 1 is 118, with 41 in Generation 1 and 37 and 39 in Generation

2 Negative/Positive cohorts respectively. The average age and female gender breakdown are 20.8/47% in Generation 1, 22.5/44% in Generation 2 Negative, and 22.3/39% in Generation 2 Positive. Subjects were paid a $5.00 show-up fee and earned an additional $7.68 on average from the investment task.

**Asset allocation experimental task.** There are three primary tasks for participants in this experiment. First, participants provide a forecast of the next period return on a risky asset (referred to as Asset B in the experiment). Second, participants allocate points between a safe asset with a 3% guaranteed return (Asset A), and the risky Asset B with a variable return. Third, participants periodically provide forward-looking belief estimates on Asset B including forecasts of the mean, maximum, minimum, and tails. We next outline these tasks, show the software interface, and provide relevant details.

1. The condition instructions were read aloud (see Appendix II in S1 File for representative condition instructions) and participants were given a $5.00 show up fee and told this was separate compensation from earnings in the investment task.

2. The first task in the experiment is for the participant to estimate the return on the risky Asset B in the next period. The software interface is shown in Fig 2. Participants were told "*the returns on Asset B will be actual real world returns from a well-known Stock Index from a specific sequential time period presented in a random order*" (see Appendix II for Experimental Instructions). In the instructions before the experiment began, participants are told the actual historical returns on the risky Asset B (S&P 500) ***prior to*** the sequential time period the participant will experience. Table 1 shows the historical return information given to the three cohorts in the experiment. For example, for Cohort 2, the actual historical return information on the S&P 500 from 1872−1950 is a mean of 4%, standard deviation of 19%, a maximum return of 46% and a minimum return of −46%. The participant then enters their forecast of the return on Asset B in the next period in the software interface shown in Fig 2. If the participant's estimate and the actual return on Asset B is within +/- 10% of each other, the participant receives a payment of 25,000 points ($0.25). The participant then advances to the asset allocation task.

3. The second task for a participant is to make the asset allocation decision. The asset allocation task requires the participant to allocate their account balance between Asset A (safe) and Asset B (risky). Participants were endowed with 100,000 points to begin the experiment and told the conversion rate for payment at the end of the experiment was $1/100,000 points. The software interface is shown in Fig 3. The participant enters the percentage portfolio allocation to

| Belief Data for Year: | Practice 2 |
|---|---|
| Last year you estimated the ACTUAL return on Asset B would be: | 10% |
| The actual return on Asset B last year was: | 0% |
| For your estimate last year you have earned points of: | 25,000 € |
| The total points earned so far: | 50,000 € |

| | Enter a number between -100 and +100 below: | |
|---|---|---|
| What do you believe the % return on Asset B will be in the next year of this experiment? YOU WILL EARN 25€ POINTS IF YOUR ESTIMATE IS WITHIN +/- 10% OF THE ACTUAL RETURN. | 5 | After filling in the boxes to the left, CLICK ON THIS BUTTON to make your next decision. |

**Fig 2. Belief estimate.**

**Table 1. Experimental design.**

| | Cohort 1 | Experiment 1 Cohort 2 | Experiment 2 Cohort 3 |
|---|---|---|---|
| Base Rate S&P 500 Data Provided in Condition Instructions* (Mean/STD/Max/Min) | 1872-1920 (2%/16%/43%/-33%) | 1872-1950 (4%/19%/46%/-46%) | 1872-1980 (5%/18%/46%/-46%) |
| Returns Experienced During Investment Task** (Mean/STD/Max/Min) | 1921-1950 (6%/22%/46%/-46%) | 1951-1980 (8%/17%/41%/-29%) | 1981-2010 (9%/17%/35%/-41%) |
| Experimental Conditions •Generational Advice Provided in Condition Instructions*** | None | Gen 1 Positive Advice | Gen 1 Positive + Gen 2 Positive |
| | | Gen 1 Negative Advice | Gen 1 Negative + Gen 2 Positive |
| | | | Gen 1 Positive + Gen 2 Negative |
| | | | Gen 1 Negative + Gen 2 Negative |

*The base rate data includes the annual price appreciation returns on the S&P 500 (excluding dividends) calculated from December to December for all years prior to the starting year of the cohort experience.

**The experienced returns are the actual returns on Asset B for the 30-year investment period.

***The generational advice is the information passed from one generation to the next in the condition instructions.

**Fig 3. Asset allocation decision.**

Asset B and the allocation to Asset A automatically sets to 100-Asset B allocation. The screen shows the total account points balance to be allocated.

4. Once a final allocation decision is made, the return information on each asset is presented as seen in Fig 4. The results page also shows the actual portfolio percentage return, the number of points gained or lost, and the new ending account balance.

5. The third task for the participant is to provide a more comprehensive estimate of the distributional properties of Asset B going forward. Prior to the first period and every ten periods thereafter, the participant is asked several questions

Click here when you have finished reviewing the results and are ready to continue.

RESULTS

| Results for Year: | % Asset A Allocation | % Asset B Allocation | Asset A % Return | Asset B % Return | Actual Porfolio % Return | Yearly Account € Return: | Ending Account Balance €: |
|---|---|---|---|---|---|---|---|
| | | | | | | | |
| 1 | 50% | 50% | 3% | 27% | 15% | 14,765 € | 114,765 € |
| 2 | 40% | 60% | 3% | 20% | 13% | 15,096 € | 129,861 € |
| 3 | 60% | 40% | 3% | -7% | -1% | -1,217 € | 128,645 € |
| | | | | | | | |
| | | | | | | | |
| | | | | | | | |

**Fig 4. Investment results.**

including a risk assessment of Asset B (*On a scale from 1 to 5 how risky is Asset B to you*), and to provide estimates of what the average, high, low, and the tails of the Asset B distribution will be going forward. The software interface is shown in in Fig 5.

6. Participants repeated these tasks for 30 years (periods). To avoid time horizon effects, participants were not told there would be 30 investment periods, but rather that they would be able to finish the task easily within the allotted time of one hour.

To summarize participant payments and incentives, participants were paid a $5.00 lab show-up fee, paid for the total points accumulated across the thirty periods in the asset allocation task, and paid for correct forecasts (within +/- 10%) on

We would now like your assesment of Asset B. Please answer the following questions:

| | Enter Answers in this Column: | Whole Number Entered should be between: |
|---|---|---|
| 1. On a scale from 1-5 (1= Not at all risky, 5= Extremely Risk) how risky is Asset B to you? | 3 | 1-5 |
| 2. Going forward, I think the **AVERAGE % return** on Asset B will be: | 10 | -100 to +100 |
| 3. Going forward, I think the **HIGHEST % return** on Asset B will be: | 42 | -100 to +100 |
| 4. Going forward, I think the **LOWEST % return** on Asset B will be: | -30 | -100 to +100 |
| 5. Going forward, I think the *percent chance* that Asset B will have a return **HIGHER THAN 30%** is: | 15 | 0 to +100 |
| 6. Going forward, I think the **percent chance** that Asset B will have a return **LOWER THAN 30%** is: | 10 | 0 to +100 |

Click on this button when you are ready to continue.

**Fig 5. Risk attitude questions.**

the annual return on the risky asset each period. Across Experiments 1 and 2 reported in this paper, participants on average received $4.61 (range from $1.63 to $15.01) for the points accumulated in the asset allocation task, and $2.82 (range from $0.00 to $6.00) for forecasts on the risky asset, for an average total payment of $12.42 (range from $8.43 to $26.01) for approximately 45 minutes of participation.

**Experimental design – Experiment 1.** The outline of the experimental design is presented in Table 1. To implement the idea of "intergenerational" advice being passed along we used the following procedure. The first cohort (Generation 1) participated in the allocation task and were given historical base rate information regarding the returns on Asset B. Generation 1 received the actual return stream on the S&P 500 from 1921–1950 although they were not told the time-period from which the returns came. The base rate information given to Generation 1 prior to beginning the experiment are the returns on the S&P 500 from 1872–1920 (see Table 1). Each Generation in the experiment is given the base rate information on the S&P 500 from 1872 to the year before the starting year of their respective return stream.

A key element of the experiment is the explanation for how the returns on Asset B would be generated. Participants were told the following in the condition instructions (see Appendix II in S1 File). Each year of the experiment the return stream on the risky asset will be the actual real world returns on a large well-known stock index in the time period from 1892–2021 drawn from a specific smaller sequential time period (e.g., 1970–2004) presented in a random order. This explanation was meant to convey three key design elements. First, the returns would be the actual returns from a real-world stock index. Second, the returns would be from a specific sequential time period. This is to convey the idea to the participant that they are experiencing a return stream an actual time period and not from a random draw from the whole history of returns. And three, presenting the returns in a random order was done to disguise the time period from which they came. The actual return streams experienced by each cohort is shown in Table 2. The S&P 500 return streams came from a data set maintained by Robert Shiller (http://www.econ.yale.edu/~shiller/data.htm). Note that the annual returns are the change in the price of the S&P 500 index from December to December, excluding dividends. The sequential year block of returns given to subjects is presented in a random order in case a participant happens to know the actual S&P 500 return stream.

At the completion of the asset allocation task, the Generation 1 participants are then directed to complete an online survey. To capture the intergenerational advice that the Generation 1 cohort will pass along to the Generation 2 cohort, the participants are told to "imagine that you are a parent and are passing advice to your children." and then the following two questions are presented:

- *Question 1: Below, please rank the following statements based on which statement you would most prefer is told to the next set of participants.*

  - *Investing in Asset B is dangerous as you can lose almost everything. For example, there was a three-year period where if you invested €100,000 in Asset B, then you would only have €40,442 left. (1)*

  - *Investing in Asset B is how you make a lot of money. For example, there was a three-year period where if you invested €100,000 in Asset B, then you would have over €241,116. (2)*

  - *Asset B is going to keep going up. Put all your money in Asset B. (3)*

  - *Asset B is going to crash. Do not put any money in Asset B. (4)*

- *Question 2: What message would you like to send along to the next cohort participating in this experiment?*

The primary reason that the advice was framed in the context of "passing advice to your children" is to emphasize that the return streams of Generation 1 and Generation 2, and later Generation 3, are from different time periods. When Generation 2 receives the advice from Generation 1, it is framed as coming from your parents. This framing was deliberate in order for each generation to clearly understand that it came from a different and earlier time period.

**Table 2. Return streams for cohorts 1, 2 and 3.**

| Generation 1 (1921–1950) | | Generation 2 (1951–1980) | | Generation 3 (1981–2010) | |
|---|---|---|---|---|---|
| Year | Return | Year | Return | Year | Return |
| 1921.12 | 0.073 | 1979.12 | 0.122 | 2009.12 | 0.265 |
| 1932.12 | −0.192 | 1975.12 | 0.322 | 2008.12 | −0.407 |
| 1949.12 | 0.089 | 1959.12 | 0.104 | 2006.12 | 0.122 |
| 1938.12 | 0.152 | 1965.12 | 0.093 | 2010.12 | 0.118 |
| 1933.12 | 0.462 | 1966.12 | −0.113 | 1981.12 | −0.073 |
| 1928.12 | 0.326 | 1971.12 | 0.101 | 1995.12 | 0.350 |
| 1943.12 | 0.206 | **1973.12** | **−0.193** | 2001.12 | −0.140 |
| 1929.12 | −0.076 | **1974.12** | **−0.292** | 1991.12 | 0.182 |
| 1939.12 | −0.025 | **1962.12** | **−0.127** | 2005.12 | 0.052 |
| 1936.12 | 0.308 | 1954.12 | 0.408 | 1990.12 | −0.057 |
| 1925.12 | 0.226 | 1961.12 | 0.263 | 2007.12 | 0.044 |
| 1950.12 | 0.194 | 1955.12 | 0.297 | 2003.12 | 0.202 |
| 1922.12 | 0.201 | 1977.12 | −0.104 | 1988.12 | 0.147 |
| 1924.12 | 0.188 | 1970.12 | −0.012 | 1998.12 | 0.237 |
| 1923.12 | −0.026 | 1958.12 | 0.326 | 1984.12 | 0.001 |
| 1942.12 | 0.087 | 1976.12 | 0.180 | 1989.12 | 0.261 |
| 1926.12 | 0.083 | 1978.12 | 0.024 | 1985.12 | 0.260 |
| 1934.12 | −0.071 | 1957.12 | −0.132 | 1994.12 | −0.023 |
| 1941.12 | −0.168 | 1953.12 | −0.046 | 2002.12 | −0.215 |
| 1930.12 | −0.275 | 1967.12 | 0.172 | 1987.12 | −0.031 |
| 1944.12 | 0.141 | 1964.12 | 0.132 | 1992.12 | 0.121 |
| 1948.12 | 0.011 | 1963.12 | 0.184 | 1999.12 | 0.201 |
| 1937.12 | −0.354 | 1980.12 | 0.238 | 1993.12 | 0.070 |
| 1927.12 | 0.294 | 1952.12 | 0.112 | 2004.12 | 0.110 |
| 1935.12 | 0.408 | 1960.12 | −0.038 | 1982.12 | 0.126 |
| 1945.12 | 0.323 | 1969.12 | −0.145 | 1997.12 | 0.295 |
| 1947.12 | −0.007 | 1972.12 | 0.185 | 1986.12 | 0.199 |
| 1940.12 | −0.149 | 1968.12 | 0.118 | 1996.12 | 0.209 |
| 1931.12 | −0.456 | 1951.12 | 0.185 | 1983.12 | 0.179 |
| 1946.12 | −0.127 | 1956.12 | 0.024 | 2000.12 | −0.068 |
| **Mean** | 0.062 | | 0.080 | | 0.091 |
| **STD** | 0.224 | | 0.172 | | 0.166 |
| **MAX** | 0.462 | | 0.408 | | 0.350 |
| **MIN** | −0.456 | | −0.292 | | −0.407 |
| **Skewness** | −0.372 | | −0.206 | | −1.044 |
| **Kurtosis** | −0.263 | | −0.562 | | 1.428 |

A sample of the advice provided from the Generation 1 cohort is presented in Table 3. Two experimental conditions are then implemented in the Generation 2 cohort, one condition received only **Positive** advice from Generation 1, and the other condition received only **Negative** advice. The advice passed came either from the positive (2) or negative (1) statement in question 1 above, and two of the actual written messages from participants in Generation 1. The advice was selected to fit the narrative of being either positive or negative investment advice.

**Table 3. Sample of advice from generation 1 cohort.**

| |
|---|
| Invest in B early on just like you would invest your IRA in common stock then slowly move over to Asset A. |
| Asset B rides a high percentage return then crashes so be careful with how you allocate your money |
| high risk but also potentially high reward |
| after halfway, be really careful on how much to put on asset b |
| every 5 year period Asset B has a signifigant fall. Be mindful and do not invest more than 10% on asset B. Sometimes it will have a great return, but it is better to be cautious than sorry. Do not compromise your money on Asset B. Play it safe. |
| You should allocate at least a small percent to asset B every time |
| No risk, no reward. Remember it is random but look at that possible trends and if your asset B is negative or below 3% not worth allocating much to it. |
| Investing is a risky decision; however., in the long-run you are getting more money and increasing your earnings, even if there is some sort of crash. |
| Be wise with your decision, and focus on consistency. Asset A is the safe option and I advise allocating most of your money on A for a steady increase in your investment. |
| Asset B can be extremely volatile, so pay much closer attention to the percentage allocated to it. |
| Nice experiment I liked it a lot! |
| Be weary, once you gained a big profit do not gamble all your profit on the next hands only lose percentage of previous profits captured. |
| Asset B is extremely risky. |
| Its all about risk a gut judgement, if you feel that its going to increase invest more. |
| It can be a large gamble to put a majority of you portfolio into asset B, but it sure is fun when you get a great percentage return. However it also really hurts to lose nearly half of your portfolio in one year. |
| Focus more on making money then trying to get within +/- 10% |
| The return can drop out of nowhere so be careful how you spend your money |

The condition instructions for the Generation 2 cohort frame the intergenerational advice from Generation 1 as follows: *"At the end of the experiment, you will be asked to respond to the following:*

- *For the next cohort that participates in this experiment, we would like to know what information you would like to pass along regarding investing in Assets A and B.*

- *The returns for Asset B for the next set of participants will be the Stock Index returns in the period after the period you just experienced. Imagine that you're a parent and are passing advice to your children about investing in Asset B.*

- *A sample of the information the prior cohort assessed important to pass along to you includes the following":*

***Negative Advice Cohort from Generation 1 to Generation 2***

◦ *"Investing in Asset B is dangerous as you can lose almost everything. For example, there was three-year period where if you invested €100,000 in Asset B, then you would only have €40,442 left."*

◦ *"It can be a large gamble to put a majority of you portfolio into asset B, but it sure is fun when you get a great percentage return. However it also really hurts to lose nearly half of your portfolio in one year."*

◦ *"Asset B is extremely risky."*

***Positive Advice Cohort from Generation 1 to Generation 2***

◦ *"Investing in Asset B is how you make a lot of money. For example, there was a three-year period where if you invested €100,000 in Asset B, then you would have over €241,116."*

◦ *"Invest in B early on just like you would invest your IRA in common stock."*

◦ *"High risk but also potentially high reward."*

Experiment 1 is thus a between subject design examining the impact of Positive and Negative intergenerational advice passed from Generation 1 to Generation 2.

There are several important elements regarding the return streams that Generation 1 and Generation 2 cohorts experienced. First, for both cohorts the risky asset B return stream experienced by the participants provided a higher average return compared to the base rate historical return. For Generation 1, the base rate average historical return from 1872–1920 is 2% and the experienced return stream from the years 1921–1950 is 6%. For Generation 2, the base rate average historical return from 1872–1950 is 4% and the experienced return stream from the years 1951–1980 is 8%. The return streams for both Generation 1 and 2 includes a reasonably large market crash; for Generation 1 the crash occurred and the end of experimental task in years 27–30 and the value of $1 in year 26 fell to $0.41 in year 30; for Generation 2 the crash occurred in the early years 7–9 of the task and the value of $1 in year 6 fell to $0.50 in year 9. The crash occurring in the last years of the task for Generation 1, a purposeful design choice, created a painful ending to the investment task that is likely to be particularly salient [29] and influential in the type of intergeneration advice that is passed.

**Experiment 1 hypotheses.** Drawing on the literature review and experimental design, we examine the following hypotheses in Experiment 1. The null hypothesis is that generational advice should have no impact. Why? First, the validity of the advice itself. For the advice of a prior generation to have validity, one must assume that the future returns on risky asset, the S&P 500, will be similar to past returns. Although many investors may believe this, the Securities and Exchange Commission requires that all investment marketing material come with a disclaimer like "Please note that past investment returns are not indicative of future returns." Future returns may not be similar to past returns because of differing economic environments in terms of GDP growth, interest rates, regulatory statutes, etc. Economic progress leads many people believe the future will be better than the past [28]. Second, the credibility of the advice giver is questionable. Participants were clearly informed that some of the advice came from prior participants, who were other peer students. There is little reason to believe that prior participants have any special knowledge of what the return stream in the next thirty-year period will be. Thus, the null hypothesis is that the advice should be ignored due to the quality of the advice and the credibility of the advice giver.

**Hypothesis 0** – The advice passed from Generation 1 to Generation 2 will have no impact on the allocations and beliefs of subjects in Generation 2.

As prior research has demonstrated, advice tends to have an impact because of the assumption that the advice giver has valuable information to share. If someone has just participated in the task I am about to engage in, then they are likely to have developed experiential knowledge that will be valuable to me. Furthermore, social norms will likely encourage a participant to follow the advice due to conformity or other psychological factors.

With regard to positive versus negative advice, we don't expect there to be a framing effect consistent with Prospect Theory [29] producing risk aversion for positive advice and risk seeking for negative advice. Although we found no prior research on the impact of experiential peer advice on investment decisions, it is reasonable to expect that positive (negative) advice will lead to more optimistic (pessimistic) beliefs regarding future returns leading to higher (lower) risk taking in portfolio allocations. These assumptions lead to the following hypotheses:

**Hypothesis 1** – Positive (negative) advice passed from Generation 1 to Generation 2 will lead to more optimistic (pessimistic) beliefs regarding the future returns on risky Asset B.

**Hypothesis 2** – Positive (negative) advice passed from Generation 1 to Generation 2 will lead to higher (lower) allocations to the risky Asset B.

**Hypothesis 3** – Negative (positive) advice passed from Generation 1 to Generation 2 will lead to a more conservative (aggressive) risk attitude measures.

## Experiment 1 results

**Mean allocations to risky asset by cohort.** Fig 6 shows the mean allocation to the risky asset by year for participants in the Positive and Negative cohorts in Generation 2. The average allocation to the risky asset (Asset B) is greater in the

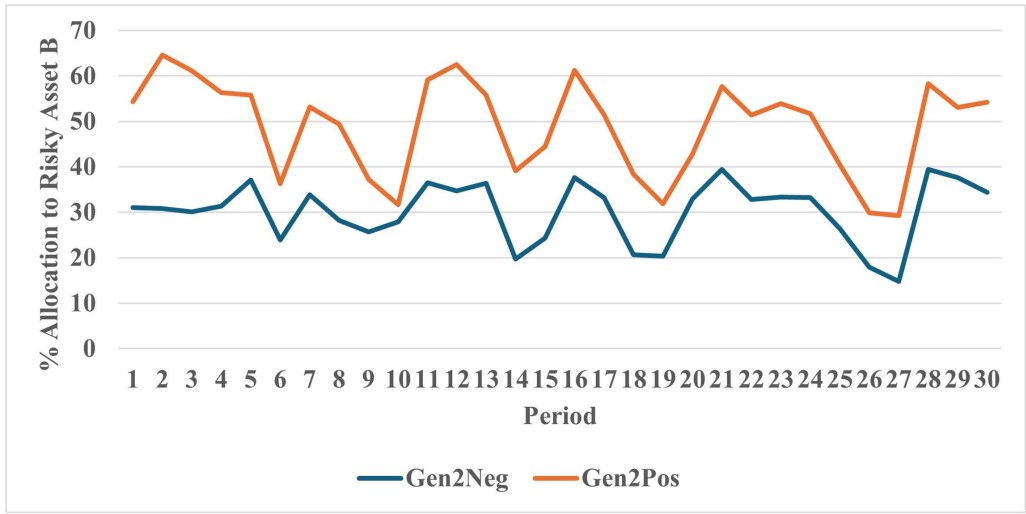

**Fig 6. Allocation to risky asset B by generation 2.**

Positive cohort compared to the Negative cohort in every year of the allocation task. Table 4 provides descriptive statistics for the mean allocations by cohort. The overall mean in allocations is 49% in the positive condition versus 30% in the negative condition. Fig 6 shows a clear and unequivocal impact of positive vs. negative advice. These visual results are confirmed in a series of statistical analyses and robustness checks (see Appendix III in S2 File). To test the robustness of the results, we conducted a number of regression specifications including OLS, Tobit, Autocorrelation, Random Effects, Multi-level, and Fractional Response models. Control variables included age, sex, education, investment experience, and lagged account balance. Results are consistent across these specifications. Estimates with controls consistently show that participants in the Negative cohort held between 13 and 15 percentage points less in stocks compared to the Positive cohort (See Appendix 3: Table A3.1 in S2 File).

Fig 7 graphs the mean beliefs of Generation 2 across the Negative and Positive cohorts. Participants in the Negative cohort held lower beliefs regarding the future returns on the risky asset in almost all thirty experimental periods. Across different regression specifications, the average of beliefs in Negative cohort is approximately 2.5 percentage points lower compared to the Positive cohort (See Appendix 3: Table A3.3 in S2 File).

To test whether differences in stock allocations across the Positive and Negative cohorts are due to differences in beliefs or risk attitudes, we conducted both regression analysis and path analysis (Appendix III and IV in S2 File). Using path analysis, we tested if beliefs or risk attitudes mediate part of the treatment effect of negative vs positive advice on stock allocations. Results show that the Generation 2 cohort receiving negative advice held lower beliefs compared to the cohort receiving positive advice and there is a statistically significant indirect effect in the model (see Table A4.1, Columns 1–4). This suggests that part of the treatment effect is driven by changes in beliefs. These changes in beliefs are associated with lower stock allocations. In contrast, the results do not find a significant indirect effect for risk attitude (see

**Table 4. Descriptive statistics, allocations and beliefs, experiment 1.**

| Treatment | Average of Risky Asset Allocation | STD of Risky Asset Allocation | Average of Belief | STD of Belief | # of Subjects |
|---|---|---|---|---|---|
| Generation 1 | 44.3 | 27.9 | 0.10 | 0.15 | 41 |
| Generation 2 Negative | 30.2 | 24.1 | 0.06 | 0.14 | 37 |
| Generation 2 Positive | 48.9 | 33.3 | 0.09 | 0.17 | 39 |

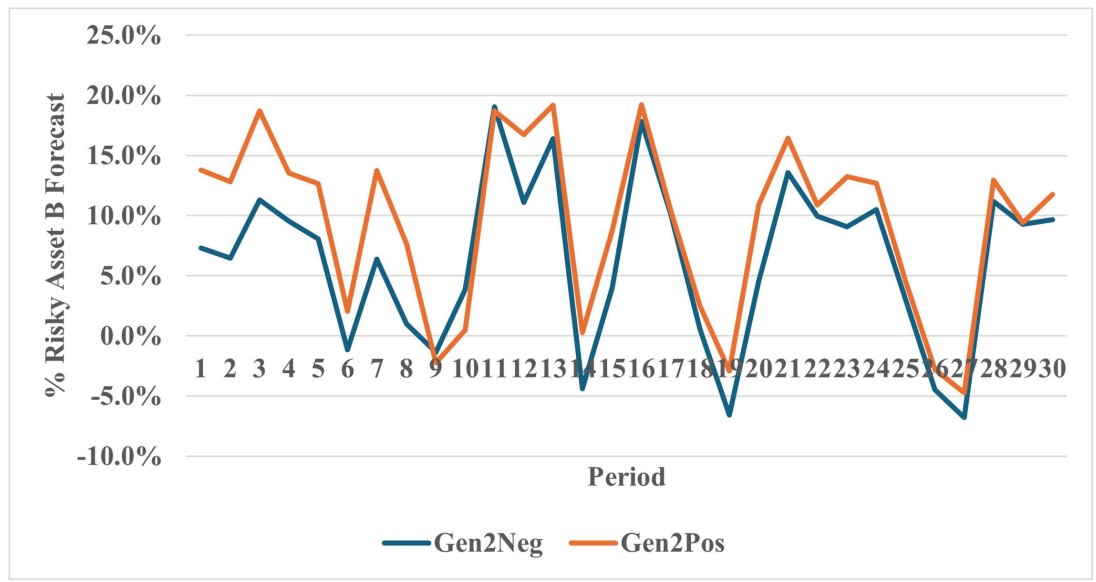

**Fig 7. Beliefs of generation 2 by treatment.**

Appendix 4, Table A4.1, Columns 6 and 8 in S2 File). That is, the effect of negative vs positive advice did not appear to significantly change risk attitudes. Taken together, the analysis for Generation 2 finds support that some of the treatment effect operated by influencing beliefs and these beliefs influenced stock choices.

## Discussion experiment 1

There are two primary findings from Experiment 1. First, participants that are given positive advice framed as coming from Generation 1 allocate more to a risky asset in all thirty years of an investment task. Second, the channel of transmission of advice to allocations appears to be through participants forming more optimistic beliefs regarding future returns than from any change in the risk attitudes.

These differences are noteworthy given the design of the experiment. Participants in both the Positive and Negative conditions were given the exact same historical base rate return information, experienced the exact same return stream over the thirty-period investment task, and faced the exact same market crash in the middle years of the task. The only difference between the two conditions was the positive or negative advice passed on from Generation 1. It was clearly explained to participants that the return stream they would be experiencing was the actual real-world return stream that occurred after the years of Generation 1.

How should a purely "rational" participant behave in this investment task? Participants in Generation 2 are told that the historical average return on the risky asset was 4% (STD 19%) and the safe asset would provide a guaranteed return of 2%. Based upon modern portfolio theory [30] participants might have selected a portfolio allocation (e.g., 60/40) and stuck with it from period to period but very few did. Only one subject out of 118 did not change the allocation at all over the thirty periods and the average standard deviation in allocations across all subjects was about 30. Prior research (7) on a similar investment task has shown that participants change their allocations quite frequently in response to recent returns on the risky asset.

Just like most naïve investors starting out on saving for retirement, participants in this experiment likely had little experience in making portfolio allocation decisions and forming beliefs about future returns. As such, participants took the advice of a prior cohort with whom they knew little about other than they had participated in a similar experiment. And

yet, subjects in Generation 2 appeared to anchor on the positive or negative advice passed on from Generation 1 and adjusted their beliefs and allocations accordingly.

Given these findings, we attempted to replicate and extend these results in a second experiment. Experiment 2 uses the same base design as Experiment 1 except now advice is passed from both Generation 1 and 2 to Generation 3.

**Experiment 2: Task, research design, participants, hypotheses**

**Asset allocation experimental task.** The asset allocation task in Experiment 2 is identical to the task in Experiment 1. As seen in Table 1, the base rate information provided to participants in Experiment 3 included the average historical returns on the S&P 500 from 1872−1980. The mean/standard deviation/maximum return/minimum return on the S&P 500 over this time period was 5%/18%/46%/-46% respectively. This base rate information is more favorable than the base rates given to Generation 1 and Generation 2 as the returns on the S&P 500 were higher later in the century. The actual return stream experienced by participants in Experiment 3 over the 30-year period provided a mean of 9% with a standard deviation of 17%; the 9% mean of this return stream is higher than those experienced by Generations 1 (6%) and 2 (8%).

**Experimental design – Experiment 2.** Experiment 2 is designed to assess the impact on Generation 3's investment decisions if they are receiving advice from both Generations 1 and 2 (see Table 1). There are four conditions in Experiment 3: 1) Generation 1 Positive Advice/Generation 2 Positive Advice; 2) Generation 1 Positive Advice/Generation 2 Negative Advice; 3) Generation 1 Negative Advice/Generation 2 Positive Advice; 4) Generation 1 Negative Advice/Generation 2 Negative Advice. The condition instructions framed the generational advice thus:

• *At the end of the experiment, you will be asked to respond to the following:*

  *"For the next cohort that participates in this experiment, we would like to know what information you would like to pass along regarding investing in Assets A and B.*

  *The returns for Asset B for the next set of participants will be the Stock Index returns in the period after the period you just experienced. Imagine that you're a parent and are passing advice to your children about investing in Asset B…"*

• *Two prior cohorts (Generation 1 and Generation 2) have completed the experiment before you. **You can think of Generation 1 as your grandparents and Generation 2 as your parents, and you are Generation 3**. Each generation experiences the actual returns on Asset B that occurred during their assigned time period.*

• *Generation 1 passed information along to Generation 2 before they began the experiment. Now Generation 1 and Generation 2 are passing information along to you (Generation 3).*

  Examples of the positive and negative advice passed from Generation 1 and 2 to Generation 3 are shown in Table 5.

**Participants experiment 2.** Participants were recruited from announcements in classes and a university subject pool. The total number of participants in Experiment 2 is 153, with 34/39/38/41 in the Positive/Positive, Positive/Negative, Negative/Positive, Negative/Negative treatments respectively. The average age in each treatment is 22.4/22.2/21.5/22.4 and the percent of females is 31%/47%/38%/45% in the four treatments respectively.

**Experiment 2 hypotheses.** Based on the results from Experiment 1, we propose the following hypotheses for Experiment 2:

**Hypothesis 4:** Positive/Positive advice from Generations 1 and 2 to Generation 3 will lead to higher allocations to the risky asset than that seen in Experiment 1. Specifically, the average allocation to the risky asset in this treatment will be significantly greater than 48.9%.

**Hypothesis 5:** Negative/Negative advice from Generations 1 and 2 to Generation 3 will lead to lower allocations to the risky asset than that seen in Experiment 1. Specifically, the average allocation to the risky asset in this treatment will be significantly less than 30.2%

**Table 5. Experiment 3 conditions.**

| Advice Passed from: | | Example of Advice |
|---|---|---|
| **Generation 1** | **Generation 2** | |
| Positive | Positive | Gen1Pos: "Invest in B early on just like you would invest your IRA in common stock." Gen2Pos: Asset B was mostly good for me. I wish I had trusted my gut more and put in more money and higher %'s. I thought it was going to be bad but it is not as risky. Mostly good. |
| Positive | Negative | Gen1Pos: "Invest in B early on just like you would invest your IRA in common stock." Gen2Neg: Be careful putting all your money in Asset B. Although it can have high returns, you don't know when it could go down and you will end up losing a lot. |
| Negative | Positive | Gen1Neg: It can be a large gamble to put a majority of your portfolio into asset B, but it sure is fun when you get a great percentage return. However it also really hurts to lose nearly half of your portfolio in one year. Gen2Pos: Asset B was mostly good for me. I wish I had trusted my gut more and put in more money and higher %'s. I thought it was going to be bad but it is not as risky. Mostly good. |
| Negative | Negative | Gen1Neg: It can be a large gamble to put a majority of your portfolio into asset B, but it sure is fun when you get a great percentage return. However it also really hurts to lose nearly half of your portfolio in one year. Gen2Neg: Be careful putting all your money in Asset B. Although it can have high returns, you don't know when it could go down and you will end up losing a lot. |

**Hypothesis 6 –** Given mixed advice, negative advice/information will have a greater impact on allocation decisions than positive advice. Given the mean allocations in the Generation 2 treatments (Positive = 48%, Negative = 30%) conditions, we expect allocations in both conditions of Generation 3 to be closer to 30% than 48%.

## Results experiment 2

**Descriptive statistics experiment 2.** The mean allocations to risky Asset B and participant beliefs in the four treatments in Experiment 2 are shown in Table 6 and Fig 8. As seen in Fig 8, the Positive/Positive (Negative/Negative) advice treatment produced the highest (lowest) allocations across all periods of the four treatments in Experiment 2. The average allocation in the Positive/Positive treatment of 59% is the highest across all six treatments in Experiments 1 and 2. The Negative/Negative treatment had the lowest average allocations (34%) in Experiment 2. To test the robustness of the results, we conducted a number of regression specifications including OLS, Tobit, Autocorrelation, Random Effects, Multi-level, and Fractional Response models. Control variables included age, sex, education, investment experience, and lagged account balance. Additional robustness checks were conducted with additional controls (See Appendix VI in S2 File). Results are consistent across these specifications.

Estimates with controls consistently show that the Negative/Negative cohort held between 17 and 23 percentage points less in stocks compared to the Positive/Positive cohort. The mixed advice conditions produced mean allocations in between the all-positive and all-negative advice treatments. Fig 9 graphs the allocation by period across all groups in

**Table 6. Descriptive statistics, allocations and beliefs, experiments 1 and 2.**

| Treatment | Average of Risky Asset Allocation | STD of Risky Asset Allocation | Average of Belief | STD of Belief | # of Subjects |
|---|---|---|---|---|---|
| Generation 1 | 44.3 | 27.9 | 0.10 | 0.15 | 41 |
| Generation 2 Negative | 30.2 | 24.1 | 0.06 | 0.14 | 37 |
| Generation 2 Positive | 48.9 | 33.4 | 0.09 | 0.17 | 39 |
| Gen3PosPos | 59.1 | 32.5 | 0.15 | 0.18 | 34 |
| Gen3PosNeg | 45.7 | 30.0 | 0.11 | 0.13 | 39 |
| Gen3NegPos | 47.0 | 29.9 | 0.13 | 0.16 | 38 |
| Gen3NegNeg | 34.0 | 27.2 | 0.13 | 0.18 | 41 |
| Grand Total | 44.17 | 29.29 | 0.11 | 0.16 | 269 |

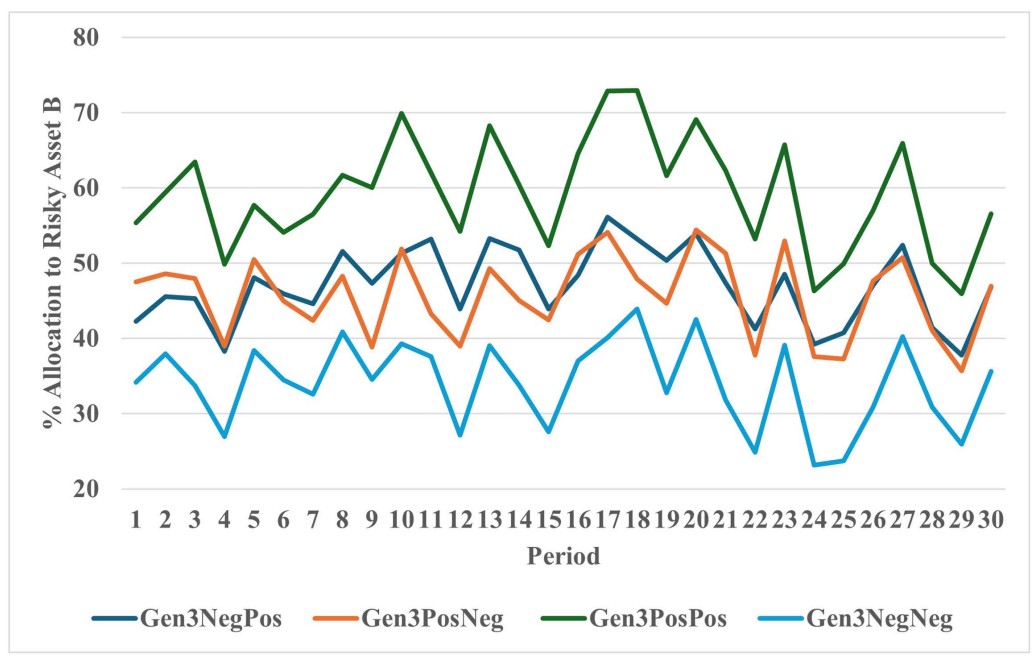

**Fig 8. Allocation to risky asset b by generation 3 by treatment.**

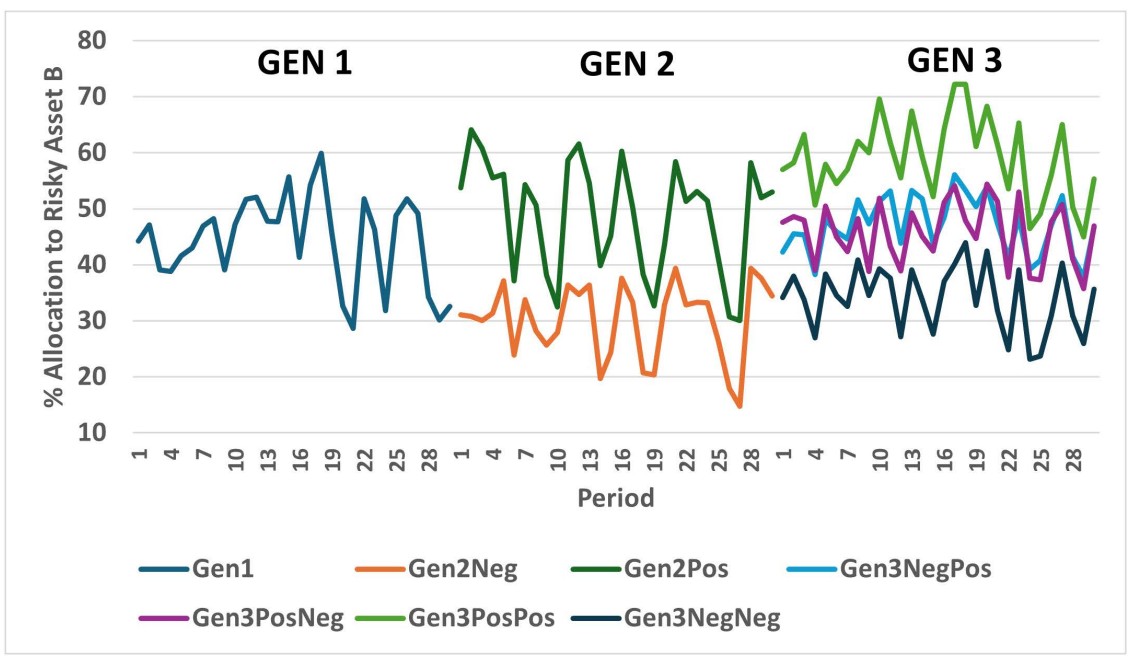

**Fig 9. Percent allocation to risky asset B by generation 1, 2 and 3.**

Generations 1, 2 and 3. The graph clearly illustrates that the Positive/Positive treatment produced the highest allocations across all three generations and treatments.

Fig 10 graphs the belief forecast held by participants in the Negative/Negative and Positive/Positive treatments across periods. Unlike in Experiment 1 where participants in the Positive condition held higher beliefs across all periods compared to subjects in the Negative condition, there is not a consistent pattern of beliefs in Experiment 2. The mean belief in the Negative/Negative treatment is 13% which is lower than the mean belief in the Positive/Positive condition of 15% but the difference is not statistically significant (see Appendix VI: Table A6.3 in S2 File).

To test whether differences in stock allocations across the Positive/Positive and Negative/Negative cohorts are due to differences in beliefs or risk attitudes, we conducted both regression analysis and path analysis (See Appendix V and IV in S2 File). Using path analysis, we tested if beliefs or risk attitudes mediate part of the treatment effect of receiving advice from multiple generations advice on stock allocations. Results show that Generation 3 with Negative/Negative advice did not hold significantly different beliefs compared to receiving positive advice and the indirect effect is not statistically significant (see Appendix IV, Table A4.2, Columns 1–4 in S2 File). This suggests that changes in beliefs do not explain the differences in allocations across cohorts. In contrast, the results do find a significant indirect effect for risk attitudes without controls (See Appendix IV, Table A4.1, Columns 6 in S2 File). The indirect effect is not significant when including controls for age, sex, education, and experience. The analysis for Generation 3 finds support that some of the treatment effects may partially be explained by changes in risk attitude.

## Discussion experiment 2

The results of Experiment 2 replicate the main results of Experiment 1 regarding asset allocations. Participants receiving positive/positive advice from prior cohorts, with the advice being framed as coming from grandparents/parents, allocate significantly more to a risky asset compared to participants receiving negative/negative advice. Both Experiments 1 and 2 suggest that there is a direct impact from advice to allocations to the risky asset.

Regarding the indirect effect of advice through the channels of transmission of beliefs and risk attitudes, Experiments 1 and 2 produce differing results. In Experiment 1 the indirect channel of transmission appears to go through beliefs,

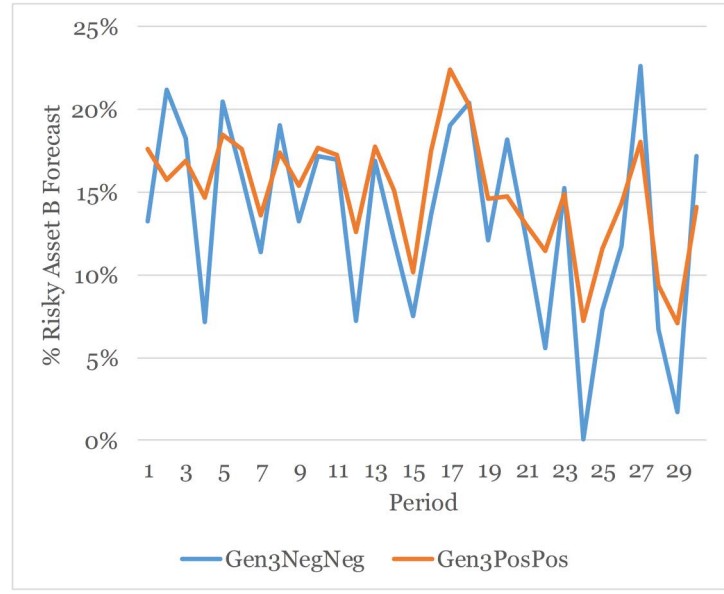

**Fig 10. Beliefs of generation 3 for positive/positive and negative/negative treatments.**

whereas in Experiment 2 the indirect channel of transmission appears to go through risk attitudes. These contrasting results suggest that further research is necessary to continue to explore the potential moderating/mediating effects of beliefs and risk attitudes.

Upon completion of the asset allocation task all participants across conditions were asked to provide a free response to explain how they formed their beliefs regarding the returns on the risky Asset B. Appendix V (see S2 File) provides a sentiment analysis of these written responses but there was no significant difference in sentiment across conditions.

## Discussion

The experimental results displayed in Fig 9 are quite clear. Participants in Generation 2 and Generation 3 that received positive advice from the prior generation(s) allocate significantly more to a risky asset compared to those receiving negative advice. This difference in allocations is apparent in the first period of the investment task and remains relatively constant over the thirty periods of the task.

There are several reasons to expect that there should not be a treatment effect. First, both groups received the exact same base rate information regarding the historical returns on the risky asset. Second, both groups are told that the returns they will experience will come from a consecutive year period after the prior generation. Third, both the positive and negative advice groups experienced the exact same thirty-period stream of returns in the exact same order during the asset allocation task. Fourth, both groups are told they are receiving advice from a prior generation who experienced the return stream preceding the return stream they would experience. Finally, both groups received the exact same multipage condition instructions except for the positive or negative advice section.

How should participants behave in the asset allocation task? Standard investment advice is to select a portfolio allocation (e.g., 60% risky, 40% safe) and stick with it. Almost none of the participants followed such advice and most adjusted their allocations from period to period as if they were trying to guess at the future return on the risky asset. This trading behavior suggests that these college-aged participants are relatively inexperienced and naïve when it comes to making the types of decisions asked of them in the experimental task. Given this inexperience, the participants are likely open to investment advice and latched on to the only advice provided, that from the prior cohort.

Why was this advice so impactful as to produce such significant differences in asset allocations? Nobel laureate Robert Shiller proposes that the field of economics has not paid sufficient attention to the importance of "narrative economics" [31]. Shiller writes:

> "The human brain has always been highly tuned toward narratives, whether factual or not, to justify ongoing actions, even such basic actions as spending and investing. Stories motivate and connect activities to deeply felt values and needs" [49, pg. 967].

And narratives are especially important for inexperienced people. Shiller continues:

> It is important to note that narratives may not be generally acted upon reflectively, since in the words of psychologists Schank and Abelson [33], they may be taken as **scripts**. When in doubt as to how to behave in an ambiguous situation, people may think back to narratives and adopt a role as if acting in a play they have seen before. The narratives have the ability to produce **social norms** that partially govern our activities, including our economic actions. The "prudent person rule" in finance is one of those norms that has economic impact: fiduciaries and experts do not have the right to act on their own judgment. They must instead mimic the "prudent person," and this, in effect, means following a script" [32], pg. 972.

What script and social norms are the subjects in this experiment likely following? Two possibilities seem likely. First, participants may have paid attention to the positive or negative content of the advice. For example, the positive advice from

Generation 1 to Generation read "*Investing in Asset B is how you make a lot of money.*" The negative advice read "*Investing in Asset B is dangerous as you can lose almost everything.*" Positive and negative framing has long been known to have differential effects on economic behavior [30]. The positive advice may have put subjects into a greed mode whereas the negative advice could have promoted a fear mode leading to higher and lower allocations respectively.

The other script that may have been activated in subjects is connected to the statement "*Imagine that you're a parent and are passing advice to your children about investing in Asset B.*" The participants in this experiment had little to no experience with the investment task they were asked to complete. What is the default script and social norm when we don't know what to do? We listen to advice, especially the advice from someone who has experience with the relevant task. Participants were given positive or negative advice framed as coming from their parents and grandparents, which likely activated social norms to follow such advice especially in an environment where most had little knowledge or experience on how to behave. Thus, a reasonable explanation for the significant experimental treatment effects is that positive or negative investment advice from a prior cohort that has experience with the task has an influential effect on your beliefs, risk aversion, and asset allocation choices.

### Limitations and future research

While this research suffers from the standard limitations of laboratory research, three issues are noteworthy. First, our experimental design does not allow us to separate out the influence of the source of the investment advice, Generation 1 and 2, from the content of the advice, positive or negative. Second, our prompt in the instructions to participants that this advice can be framed as coming from parents and grandparents is rather mild and thus should not be construed as actual parental advice Third, our ability to clearly delineate the causal impact of advice through the transmission channels of beliefs or risk aversion is limited by our ability to accurately measure beliefs and risk attitudes in the laboratory. Future experimental designs should attempt to overcome all these limitations.

Future research should explore the importance of the source and content of the advice as part of the narrative. Does the impact of the advice depend upon whether it comes from parents and grandparents, peers, financial experts, friends, etc.? How does the content of the advice matter in terms of valence, directive content, or source credibility matter? This laboratory research should be complemented with field studies examining what type of financial advice, if any, parents do pass along to children, and a longitudinal analysis of how such advice impacts a person's lifetime investment choices. Finally, laboratory interventions should be explored to reduce the amount of variance in participants allocations from period to period which almost certainly leads to lower investment performance.

### Supporting information

**S1 File. Consent form and condition instructions.**
(DOCX)

**S2 File. Supplemental data analyses.**
(DOCX)

### Author contributions

**Conceptualization:** Garret Ridinger, James Sundali, Federico Guerrero, Mauricio Solorio, Mengyue Fan, Irem Sevindik, Qifan Chen, Diana Neely.

**Data curation:** James Sundali, Mauricio Solorio, Mengyue Fan, Irem Sevindik, Qifan Chen, Diana Neely.

**Formal analysis:** Garret Ridinger, Federico Guerrero, Mauricio Solorio, Mengyue Fan, Irem Sevindik.

**Methodology:** Garret Ridinger, James Sundali, Federico Guerrero, Mauricio Solorio, Mengyue Fan, Qifan Chen, Diana Neely.

**Project administration:** James Sundali.

**Software:** Mengyue Fan.

**Supervision:** James Sundali, Federico Guerrero.

**Writing – original draft:** Garret Ridinger, James Sundali, Federico Guerrero, Irem Sevindik.

**Writing – review & editing:** Garret Ridinger, James Sundali, Federico Guerrero, Mengyue Fan, Irem Sevindik, Qifan Chen, Diana Neely.

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
