## [Decision Letter · Decision Letter 0]

5 Nov 2025

Dear Dr. Sundali,

Thank you for submitting your manuscript to PLOS ONE. After careful consideration, we feel that it has merit but does not fully meet PLOS ONE’s publication criteria as it currently stands. Therefore, we invite you to submit a revised version of the manuscript that addresses the points raised during the review process.

We look forward to receiving your revised manuscript.

Kind regards,

Alexander Zimper

Academic Editor

PLOS ONE

Journal Requirements:

4. Please include captions for your Supporting Information files at the end of your manuscript, and update any in-text citations to match accordingly. Please see our Supporting Information guidelines for more information: http://journals.plos.org/plosone/s/supporting-information .

5. We note that there is identifying data in the Supporting Information file < Appendix.docx>. Due to the inclusion of these potentially identifying data, we have removed this file from your file inventory. Prior to sharing human research participant data, authors should consult with an ethics committee to ensure data are shared in accordance with participant consent and all applicable local laws.

-Location data

Additional Editor Comments:

Dear Authors,

I have now received two reports. Reviewer 1 is rather positive and makes mainly editorial suggestions. Reviewer 2 is quite negative and provides many detailed critique points. His/her main concerns are (i) that you are too imprecise in arriving at conclusions from your experiment and (ii) that you are overselling some of your results. Both reviewers agree that your findings are not surprising.

Although Reviewer 2 recommends rejection, he/she writes to me:

"If you decide to help the authors to publish this paper, there will be a substantial revision, rewriting many sessions and tone down the story of intergeneration advice. They can talk about social influence instead."

I am willing to give you the opportunity to substantially revise the paper in order to make it more precise and toned-down.

Good luck,

Alex

Reviewer's Responses to Questions

**Comments to the Author**

1. Is the manuscript technically sound, and do the data support the conclusions?

Reviewer #1: Partly

Reviewer #2: No

2. Has the statistical analysis been performed appropriately and rigorously?

Reviewer #1: Yes

Reviewer #2: Yes

3. Have the authors made all data underlying the findings in their manuscript fully available?

Reviewer #1: Yes

Reviewer #2: No

4. Is the manuscript presented in an intelligible fashion and written in standard English?

Reviewer #1: Yes

Reviewer #2: No

Reviewer #1: The contribution is incremental and effectively highlights the effects of counseling. While well-conducted, the findings may not significantly alter the views of those familiar with social learning theory, as they may be seen as extensions of existing ideas.

Here are some recommendations:

1) The experimental design is clear and valid; however, causality claims should be made cautiously, as real-world connections require more justification.

2) Framing the treatment as "parental advice" may lead to conformity with social norms rather than genuine belief changes; more precise distinctions between "beliefs" and "risk aversion" are needed.

3) Overall, the writing is engaging, but the term "significant impact" could be clarified; distinguishing between economic and statistical significance is important.

4) Consider reviewing the formatting of figures that resemble spreadsheet tables.

5) Adding a conclusions section is suggested, though optional.

These comments may benefit this research.

Reviewer #2: The paper studies whether “intergenerational” investment advice, messages framed as coming from a prior cohort (framed as “parents”/“grandparents”), affects subsequent cohorts’ portfolio choices and beliefs in a 30‑period asset‑allocation task. In Experiment 1, Generation 2 (G2) is randomly assigned to receive either positive or negative advice written by investors in Generation 1 (G1). All G2 subjects then face the same return stream (S&P 500 price returns from 1951–1980 in randomized order, though I have a question about this, see below), allocate between a safe asset and a risky asset each period, and provide point forecasts of the next risky return. Experiment 2 gives Generation 3 (G3) two pieces of advice (from G1 and G2), yielding four conditions thanks to the mixture of experiences: Pos/Pos, Pos/Neg, Neg/Pos, Neg/Neg. The authors find that advice valence moves average risky shares materially (e.g., in G2 about 49% vs. 30% risky; in G3 about 59% in Pos/Pos vs. 34% in Neg/Neg), with differences present from the first decision and persistent over 30 periods (see Figs. 6, 8, 9; Tables 4, 6). The paper also explores mechanisms using mediation analyses, concluding that advice mainly shifts beliefs in G2 but (tentatively) risk aversion in G3. While the results look quite clear, I have some serious reservations about the quality of the paper. At its current state, it is not recommended to publish it.

Major comments

1. The core claim of “intergenerational” effects is not supported by the design. Advice comes from anonymous prior lab cohorts that are merely framed as “parents/grandparents,” not from actual parent–child links or documented family transmission. The welfare of their “children” is not explicitly in their utility function (maybe they do, but you lose control over how different people treat their children’s welfare). This is similar to social influence, not intergenerational transmission. Thus, the interpretation seems to be too much of a stretch. I would stick to a social influence story, from one cohort to the next, and certainly not sell it as an inter-generational story.

2. The mechanism claims are not credible. “Risk aversion” is proxied by a single item: “How risky is Asset B to you? (1 --- 5)”, which is risk perception, not preference. Mediation analyses then label this as a preference channel. If you measured it separately, then I must have completely missed it.

3. How exactly people were affected by the previous cohort’s message is also not entirely clear. There are pre-scripted messages but there are also free-formed messages, which carries more weight is unclear. “positive” vs. “negative” snippets differ in content, directive strength, and length and are bundled with a strong authority cue (“parents/grandparents”). The design cannot identify whether effects come from advice valence, directive content, or source credibility.

4. It is entirely unclear how you incentivize your subjects. Should we be concerned about hedging due to the payment of both belief elicitation and investment? Please clarify the payoff is calculated from portfolio outcomes to money. The text states a $5 show up fee plus “earnings in the investment task,” but the conversion for allocation payoffs is not spelled out in the main text. If allocation payoffs were small or hypothetical, demand effects from advice become a more plausible driver. Please provide more lab details. How much do your subjects earn on average?

5. You should discuss your hypotheses section more seriously, not just loosely saying that these were derived from the literature. How? It is clear to the reader.

6. Random or not? You said the returns are drawn randomly, but then you also wrote that it was your design choice to have a crash in the end to create a salient effect. Are you deceiving your subjects in the instructions, or am I missing something?

7. Presentation is not at a publishable standard: You should write the experimental design section properly, instead of copying context from the instructions with bullet points. Other minor issues include inconsistent currency symbols (€ vs $, I thought the experiment was in the US?), many typos (e.g., “Kahnneman”, “advicd”), inconsistent safe asset return, etc.

**Do you want your identity to be public for this peer review?** For information about this choice, including consent withdrawal, please see our Privacy Policy

Reviewer #1: No

Reviewer #2: No

---

## [Author Response · Author response to Decision Letter 1]

19 Dec 2025

We have followed PLOS ONE’S style and naming requirements, please let us know if there is anything we have missed.

2. When completing the data availability statement of the submission form, you indicated that you will make your data available on acceptance. We strongly recommend all authors decide on a data sharing plan before acceptance, as the process can be lengthy and hold up publication timelines. Please note that, though access restrictions are acceptable now, your entire data will need to be made freely accessible if your manuscript is accepted for publication.

We have made our data available at Science Data Bank

https://www.scidb.cn/en/s/IvI7Nv

https://www.scidb.cn/en/anonymous/SXZJN052

3. Ethics statements – We have updated the ethics statement in the Methods section as follows:

a. “This experiment, and all subsequent experiments reported in this paper, began with participants reading and signing an informed consent form that was approved by the University of Nevada, Reno Social Behavioral Institutional Review Board (Protocol # 1128451-1) (see Appendix 1 in S1 file for consent form) (IRB approval was granted on 10/06/2017 and reaffirmed on 10/26/2021).”

4. Please include captions for your Supporting Information files at the end of your manuscript, and update any in-text citations to match accordingly.

a. These supporting files have been added/titled

i. S1 Consent Form and Condition Instructions

ii. S2 Supplemental Data Analyses

5. TBD We note that there is identifying data in the Supporting Information file < Appendix.docx>. Due to the inclusion of these potentially identifying data, we have removed this file from your file inventory. Prior to sharing human research participant data, authors should consult with an ethics committee to ensure data are shared in accordance with participant consent and all applicable local laws.

We have removed the identifying information in the IRB consent form in the supporting information file.

Reviewer 1 Comments

Here are some recommendations:

1) The experimental design is clear and valid; however, causality claims should be made cautiously, as real-world connections require more justification.

Response: We thank the reviewer for this comment, and it aligns with the comments from Reviewer 2. We have substantial changed the “intergenerational story” and have replaced that phrasing with “experiential advice.” We have rewritten the literature review section and have removed the depression babies literature and a significant amount of the intergenerational literature. We are careful in our discussion of not claiming a real-world intergenerational effect. We do keep the Generation 1/Generation 2 terminology for several reasons. First, in our condition instructions to participants the advice that is being passed between cohort is referred to as advice from a parent to child. Second, in the experimental literature, information passed from one cohort to another is often referred to as generational advice. To make this analogy clear, we write in the Introduction Section of the paper:

• “The focus of the research presented here concerns the impact of experiential peer advice, often referred to as “generational advice” in the experimental literature (43, 45). We will use the term “generational advice” to be consistent with prior research and to make it clear that advice is coming from a prior experienced cohort. We are not claiming this equates to actual advice passed between family generations (i.e. parent to child).

And some of the relevant references are:

• Schotter, A. and Sopher, B., 2006. Trust and trustworthiness in games: An experimental study of intergenerational advice. Experimental Economics, 9, pp.123-145.

• Sherstyuk, K., Tarui, N., Ravago, M. L. V., & Saijo, T. (2016). Intergenerational games with dynamic externalities and climate change experiments. Journal of the association of environmental and resource economists, 3(2), 247-281.

• Brau, J. C., Holmes, A. L., & Israelsen, C. L. (2019). Financial literacy among college students. Journal of Financial Education, 45(2), 179-205.

In the literature section we write:

• “A study closely related to ours is Alevy and Price [47], who examine the role of intergenerational advice on experimental asset markets. Again, “intergenerational advice” is a proxy term for advice passed from one experienced cohort the next.”

• Alevy, J. E., & Price, M. K. (2017). Advice in the marketplace: a laboratory study. Experimental Economics, 20, 156-180.

2) Framing the treatment as "parental advice" may lead to conformity with social norms rather than genuine belief changes; more precise distinctions between "beliefs" and "risk aversion" are needed.

We again thank the reviewer for this comment. As stated above, we have moved away from framing the treatments as direct “parental advice” and make clear that advice is being passed from one experimental cohort (generation) to the next.

Since we are not directly measuring participant “risk aversion” we have replaced this term with “risk attitude” as suggest by Reviewer 2. We add the following sentence in the introduction to make clear this change and how we are measuring these variables:

• “Investment beliefs are measured as a participant’s forecast of the return on the risky asset in the next period. The participant’s risk attitude regarding the risky asset is measured from a survey question answered by the participant once every ten periods.”

3) Overall, the writing is engaging, but the term "significant impact" could be clarified; distinguishing between economic and statistical significance is important.

We have adjusted the writing to make clear we are generally referring to statistical significance.

4) Consider reviewing the formatting of figures that resemble spreadsheet tables.

Tables and figures have been reformatted to fit Plos One requirements.

5) Adding a conclusions section is suggested, though optional.

We thank the reviewer for this comment. Given the length of the paper and the summary comments in the final Discussion section of the paper, we have chosen not to add a conclusion section.

Reviewer 2 Comments

1. The core claim of “intergenerational” effects is not supported by the design. Advice comes from anonymous prior lab cohorts that are merely framed as “parents/grandparents,” not from actual parent–child links or documented family transmission. The welfare of their “children” is not explicitly in their utility function (maybe they do, but you lose control over how different people treat their children’s welfare). This is similar to social influence, not intergenerational transmission. Thus, the interpretation seems to be too much of a stretch. I would stick to a social influence story, from one cohort to the next, and certainly not sell it as an inter-generational story.

Response: We thank the reviewer for this comment, and it aligns with the comments from Reviewer 1. We have substantial changed the “intergenerational story” and have replaced this phrasing with “experiential advice.” We have rewritten the literature review section and have removed the depression babies literature and a significant amount of the intergenerational literature. We are careful in our discussion of not claiming a real-world intergenerational effect. We do keep the Generation 1/Generation 2 terminology for several reasons. First, in our condition instructions to participants the advice that is being passed between cohort is referred to as advice from a parent to child. Second, in the experimental literature, information passed from one cohort to another is often referred to as generational advice. To make this analogy clear, we write in the Introduction Section of the paper:

• “The focus of the research presented here concerns the impact of experiential peer advice, often referred to as “generational advice” in the experimental literature (43, 45). We will use the term “generational advice” to be consistent with prior research and to make it clear that advice is coming from a prior experienced cohort. We are not claiming this equates to actual advice passed between family generations (i.e. parent to child).

And some of the relevant references are:

• Schotter, A. and Sopher, B., 2006. Trust and trustworthiness in games: An experimental study of intergenerational advice. Experimental Economics, 9, pp.123-145.

• Sherstyuk, K., Tarui, N., Ravago, M. L. V., & Saijo, T. (2016). Intergenerational games with dynamic externalities and climate change experiments. Journal of the association of environmental and resource economists, 3(2), 247-281.

• Brau, J. C., Holmes, A. L., & Israelsen, C. L. (2019). Financial literacy among college students. Journal of Financial Education, 45(2), 179-205.

In the literature section we write:

• “A study closely related to ours is Alevy and Price [47], who examine the role of intergenerational advice on experimental asset markets. Again, “intergenerational advice” is a proxy term for advice passed from one experienced cohort the next.”

• Alevy, J. E., & Price, M. K. (2017). Advice in the marketplace: a laboratory study. Experimental Economics, 20, 156-180.

2. The mechanism claims are not credible. “Risk aversion” is proxied by a single item: “How risky is Asset B to you? (1 --- 5)”, which is risk perception, not preference. Mediation analyses then label this as a preference channel. If you measured it separately, then I must have completely missed it.

We thank that reviewer for this comment and concur that we are not measuring risk aversion. We have changed the language throughout the paper from “risk aversion” to “risk attitude.”

2. How exactly people were affected by the previous cohort’s message is also not entirely clear. There are pre-scripted messages but there are also free-formed messages, which carries more weight is unclear. “positive” vs. “negative” snippets differ in content, directive strength, and length and are bundled with a strong authority cue (“parents/grandparents”). The design cannot identify whether effects come from advice valence, directive content, or source credibility.

We thank the reviewer for this comment and completely agree that the design of the current experiment cannot disentangle the effects of advice valence, directive content, or source credibility. Our experiment was primarily designed to assess whether there would be a main effect of advice passed from one cohort to another, and if so, would it matter if the advice given was positive or negative? Now that we have established that experiential advice does have a clear and statistically significant effect, follow-up experiments are necessary to determine the underlying mechanisms behind the advice such as those suggested.

We have updated the limitations and future research section with the following:

• “Future research should explore the importance of the source and content of the advice as part of the narrative. Does the impact of the advice depend upon whether it comes from parents and grandparents, peers, financial experts, friends, etc.? How does the content of the advice matter in terms of valence, directive content, or source credibility matter.”

3. It is entirely unclear how you incentivize your subjects. Should we be concerned about hedging due to the payment of both belief elicitation and investment? Please clarify the payoff is calculated from portfolio outcomes to money. The text states a $5 show up fee plus “earnings in the investment task,” but the conversion for allocation payoffs is not spelled out in the main text. If allocation payoffs were small or hypothetical, demand effects from advice become a more plausible driver. Please provide more lab details. How much do your subjects earn on average?

We thank the reviewer for noting our lack of clarity. We have updated the Experimental Design instructions with the following:

• “Participants were endowed with 100,000 points to begin the experiment and told the conversion rate for payment at the end of the experiment was $1/100,000 points.”

• “To summarize participant payments and incentives, participants were paid a $5.00 lab show-up fee, paid for the total points accumulated across the thirty periods in the asset allocation task, and paid for correct forecasts (within +/- 10%) on the annual return on the risky asset each period. Across Experiments 1 and 2 reported in this paper, participants on average received $4.61 (range from $1.63 to $15.01) for the points accumulated in the asset allocation task, and $2.82 (range from $0.00 to $6.00) for forecasts on the risky asset, for an average total payment of $12.42 (range from $8.43 to $26.01) for approximate 45 minutes of participation.”

Given these payment amounts, we don’t believe hedging is a problem.

4. You should discuss your hypotheses section more seriously, not just loosely saying that these were derived from the literature. How? It is clear to the reader.

Again, we thank the reviewer for pointing out our lack of clarity. We have expanded our development of the hypotheses with the following:

“The null hypothesis is that generational advice should have no impact. Why? First, the credibility of the advice itself. For the advice of a prior generation to have validity, one must assume that the future returns on risky asset, the S&P 500, will be similar to past returns. Although many investors may believe this, the Securities and Exchange Commission requires that all investment marketing material come with a disclaimer like “Please note that past investment returns are not indicative of future returns.” Future returns may not be similar to past returns because of differing economic environments in terms of GDP growth, interest rates, regulatory statutes, etc. Economic progress leads many people to believe the future will be better than the past (27). Second, the credibility of the advice giver is questionable. Participants were clearly informed that some of the advice came from prior participants who were other students like themselves. There is little reason to believe that prior participants have any special knowledge of what the return stream in the next thirty-year period will be. Thus, the null hypothesis is that the advice should be ignored due to the quality of the advice and the credibility of the advice giver.”

“As prior research has demonstrated, advice tends to have an impact because of the assumption that the advice giver has valuable information to share. If someone has just participated in the task I am about to engage in, then they are likely to have developed experiential knowledge that will be valuable to me. Furthermore, social norms will likely encourage a participant to follow the advice due to conformity or other psychological factors.

With regard to positive versus negative advice, we don’t expect there to be a framing effect consistent with Prospect Theory [52] producing risk aversion for positive advice and risk seeking for negative advice. Although we found no prior research on the impact of experiential peer advice on investment decisions, it is reasonable to expect that positive (negative) advice will lead to more optimistic (pessimistic) beliefs regarding future returns leading to higher (lower) risk taking in portfolio allocations. These assumptions lead to the following hypotheses:”

5. Random or not? You said the returns are drawn randomly, but then you also wrote that it was your design choice to have a crash in the end to create a salient effect. Are you deceiving your subjects in the instructions, or am I missing something?

We thank the reviewer for raising the question of deception. We have been running experiments of this type for several years and different disciplines (economics vs. psychology) and journals have very different viewpoints on what constitutes deception and when, or if, deception crosses a boundary. We have tried our best to avoid ANY deception, but we acknowledge that Reviewer 2 may see it differently. Let us explain.

The instructions given to Participants were the following:

• Each year of the experiment a random draw from a distribution of the

---

## [Editor Report · Decision Letter 1]

2 Jan 2026

Dear Dr. Sundali,

Thank you for submitting your manuscript to PLOS ONE. After careful consideration, we feel that it has merit but does not fully meet PLOS ONE’s publication criteria as it currently stands. Therefore, we invite you to submit a revised version of the manuscript that addresses the points raised during the review process.

We look forward to receiving your revised manuscript.

Kind regards,

Alexander Zimper

Academic Editor

PLOS One

Journal Requirements:

Additional Editor Comments:

I am willing to accept the current version of the paper as you convincingly addressed the reviewers' and my concerns. However, please carefully recheck the paper (+ supplemental part) for spelling errors. For example, I picked up "S2 Supplemnetal...

---

## [Author Response · Author response to Decision Letter 2]

30 Jan 2026

Review Comment: I am willing to accept the current version of the paper as you convincingly addressed the reviewers' and my concerns. However, please carefully recheck the paper (+ supplemental part) for spelling errors. For example, I picked up "S2 Supplemnetal...

• Response: We thank the editor for the careful reading of our paper. We have rechecked and corrected the paper and supplements S1 and S2 for spelling errors and edits.

---

## [Editor Report · Decision Letter 2]

2 Feb 2026

An Experiment on the Impacts of Experiential Investment Advice

PONE-D-25-20919R2

Dear Dr. Sundali,

We’re pleased to inform you that your manuscript has been judged scientifically suitable for publication and will be formally accepted for publication once it meets all outstanding technical requirements.

Kind regards,

Alexander Zimper

Academic Editor

PLOS One
---

## [Editor Report · Acceptance letter]

PONE-D-25-20919R2

PLOS One

Dear Dr. Sundali,

I'm pleased to inform you that your manuscript has been deemed suitable for publication in PLOS One. Congratulations! Your manuscript is now being handed over to our production team.

Kind regards,

on behalf of

Dr. Alexander Zimper

Academic Editor

PLOS One